# Winter Holidays and Their Impact on Eating Behavior—A Systematic Review

**DOI:** 10.3390/nu15194201

**Published:** 2023-09-28

**Authors:** Irina Mihaela Abdulan, Gabriela Popescu, Alexandra Maștaleru, Andra Oancea, Alexandru Dan Costache, Doina-Clementina Cojocaru, Carmen-Marinela Cumpăt, Bogdan Mihnea Ciuntu, Bogdan Rusu, Maria Magdalena Leon

**Affiliations:** 1Department of Medical Specialties I, “Grigore T. Popa” University of Medicine and Pharmacy, 700115 Iasi, Romania; irina.abdulan@yahoo.com (I.M.A.); adcostache@yahoo.com (A.D.C.); clementina.cojocaru@gmail.com (D.-C.C.); marinela.cumpat@umfiasi.ro (C.-M.C.); leon_mariamagdalena@yahoo.com (M.M.L.); 2Clinical Rehabilitation Hospital, 700661 Iasi, Romania; popescu_gabriela96@yahoo.com; 3Department of General Surgery, “Grigore T. Popa” University of Medicine and Pharmacy, Universitatii Street, No. 16, 700115 Iasi, Romania; bogdan-mihnea.ciuntu@umfiasi.ro; 4Faculty of Industrial Design and Business Management, “Gheorghe Asachi” Technical University of Iași, 700050 Iasi, Romania; bogdan.rusu@academic.tuiasi.ro

**Keywords:** overeating, holidays, obesity, weight loss

## Abstract

(1) Background: There has been a growing interest in understanding the causes of obesity and developing effective prevention strategies. Lifestyle change programs are often considered the gold standard for weight reduction, and they can help individuals with obesity achieve an annual weight loss of around 8–10%. The aim of this review was to evaluate the effect of food during the winter holidays. This knowledge will serve as a valuable foundation for the development of targeted interventions and prevention programs. (2) Methods: We conducted a systematic search of the literature via one database (PubMed). The search was limited to studies published in English in the last 10 years, with adult participants, but without specifying limits regarding the study design. We excluded articles that addressed intermittent fasting diets or weight loss intervention methods during the holidays through various diets. (3) In separate sections, we analyzed the psychological causes of gaining weight during the winter holidays, behavioral patterns, prevention strategies and the nutritional composition of the different types of food served during the festive period. Results: Using the combination of the terms “holiday and obesity”, “holiday and weight gain”, “festive season and obesity”, and “festive season and weight gain” we obtained 216 results involving the addressed topic. Thus, only ten articles remained after screening, with a total of 4627 participants. Most participants experienced weight fluctuations during the study period, particularly during holidays. One concerning observation was that most of the weight gained during these periods was maintained even after the end of the studies, especially in those with obesity. A supervised exercise program and a controlled diet at work over the Christmas period are effective strategies for avoiding weight gain and its deleterious effects in people with metabolic syndrome or weight problems. (4) In addition, attention must be focused on the psycho-social factors during the holidays because for some people it is a stressful period and can cause a much higher caloric consumption. The simplest method to approach during the holidays is to implement small tips and tricks during this period that will prevent individuals from gaining extra pounds. Conclusions: It is essential to acknowledge that obesity is a multifaceted condition that requires a comprehensive and multidisciplinary approach to address its underlying factors and provide ongoing assistance to individuals in their weight-management endeavors. Even the most effective short-term interventions are likely to produce continued positive outcomes with persistent intervention and support.

## 1. Introduction

### 1.1. Background/Rationale

Obesity poses a significant challenge to public health, with its prevalence rapidly rising globally. It is currently the fifth leading cause of death worldwide [1].

In Europe, the prevalence of overweight and obesity among adults is 34.8% and 12.8%, respectively [2]. Even more alarming is the fact that over the past few decades, the occurrence of obesity has tripled, resulting in over two-thirds (70.2%) of the adult population in the United States being overweight or obese. Additionally, nearly half of adults (48.5%) are affected by prediabetes or diabetes, conditions strongly associated with obesity. Unfortunately, young adults are the most affected ones [3]. 

There has been a growing interest in understanding the causes of obesity and developing effective prevention strategies. Lifestyle change programs are often considered the gold standard for weight reduction, and they can help individuals with obesity achieve an annual weight loss of around 8–10% [4,5,6]. 

While these programs can initially lead to significant weight loss, sustaining this over time remains challenging for many individuals. Various factors, such as genetic predisposition, physiological changes, and environmental influences, can contribute to weight regain. This makes weight maintenance challenging and often leads to weight regain over time [7]. 

Incorporating lifestyle modifications, including maintaining a balanced diet, participating in regular physical activity, and making sustainable behavior changes, is crucial for preventing weight gain and promoting a healthy weight. These aspects should be seamlessly integrated into daily routines and maintained over the long term to achieve lasting results. When individuals are unable to achieve and sustain significant weight loss over an extended period, it is often attributed to their perceived failure to adhere to the recommended lifestyle changes. This attribution can potentially perpetuate the stigma surrounding the patient, implying a lack of willpower, motivation, or determination to lose weight. It is common for individuals to focus more on what they have not achieved rather than acknowledging their accomplishments. Unlike weight loss, where visible progress on the scale and improvements in clinical measures can boost motivation, the maintenance phase often lacks these explicit rewards. To support motivation and reinforce satisfaction with outcomes, it is important to draw attention to patients’ progress, which can sometimes be overlooked [8].

Even when the patient complies with the proposed strategy, there is a tendency to abandon the rules of a balanced diet during vacations or holidays. These periods are associated with relaxation, indulgence, and a break from regular routines, leading to unhealthy eating habits, reduced physical activity, and other behaviors contributing to weight gain. The period encompassing the last week of November and the first or second week of January poses a significant risk, as people worldwide engage in celebrations such as Christmas, New Year, and various social gatherings where high-calorie foods are consumed. These include desserts, sugary drinks, sweets, and alcohol. Moreover, during this period, physical activity tends to decrease. 

According to a study published in the New England Journal of Medicine, the average American gains just under 1 pound during the holiday season. While this might not seem significant, research indicates that this weight is often not lost as the seasons change. In fact, it can account for more than 50% of the total weight gained throughout the year [9]. When holiday weight gain is not reversed, it can contribute to a cycle of gradually putting on extra pounds over a person’s lifetime. This can increase the risk of various diseases, including diabetes and heart disease.

We consider this systematic review of valuable importance due to its extensive research, and, as far as we know, it is the first to evaluate the importance of nutritional patterns during the holidays.

### 1.2. Objectives

In the current world context, it is crucial to develop effective strategies for preventing obesity since it is challenging to reverse once it becomes established. An essential step in developing these strategies is gaining an understanding of the specific periods in a person’s life cycle that are particularly vulnerable to weight gain. 

In particular, the aim of this review was to evaluate the effect of food during the winter holidays. This knowledge will serve as a valuable foundation for developing targeted interventions and prevention programs.

## 2. Materials and Methods

### 2.1. Electronic Search Strategy

Considering these aspects, a search was conducted in PubMed using the keywords “holiday/holidays”, “weight gain”, “festive season” and “obesity”. 

1. (holiday) AND (obesity),,in the last 10 years”,((““holidaying”“[All Fields] OR ““holidays”“[MeSH Terms] OR ““holidays”“[All Fields] OR ““holiday”“[All Fields] OR ““vacation”“[All Fields] OR ““vacationed”“[All Fields] OR ““vacationing”“[All Fields] OR ““vacations”“[All Fields]) AND (““obeses”“[All Fields] OR ““obesity”“[MeSH Terms] OR ““obesity”“[All Fields] OR ““obese”“[All Fields] OR ““obesities”“[All Fields] OR ““obesity s”“[All Fields])) AND (y_10[Filter])”.

2. (festive season) AND (obesity),,in the last 10 years”, ((““festive”“[All Fields] OR ““festivities”“[All Fields] OR ““festivity”“[All Fields] OR ““holidays”“[MeSH Terms] OR ““holidays”“[All Fields] OR ““festival”“[All Fields] OR ““festivals”“[All Fields]) AND (““season s”“[All Fields] OR ““seasonability”“[All Fields] OR ““seasonable”“[All Fields] OR ““seasonably”“[All Fields] OR ““seasonal”“[All Fields] OR ““seasonalities”“[All Fields] OR ““seasonality”“[All Fields] OR ““seasonally”“[All Fields] OR ““seasonals”“[All Fields] OR ““seasons”“[MeSH Terms] OR ““seasons”“[All Fields] OR ““season”“[All Fields]) AND (““obeses”“[All Fields] OR ““obesity”“[MeSH Terms] OR ““obesity”“[All Fields] OR ““obese”“[All Fields] OR ““obesities”“[All Fields] OR ““obesity s”“[All Fields])) AND (y_10[Filter])”.

3. (holiday) AND (weight gain),,in the last 10 years”, ((““holidaying”“[All Fields] OR ““holidays”“[MeSH Terms] OR ““holidays”“[All Fields] OR ““holiday”“[All Fields] OR ““vacation”“[All Fields] OR ““vacationed”“[All Fields] OR ““vacationing”“[All Fields] OR ““vacations”“[All Fields]) AND (““weight gain”“[MeSH Terms] OR (““weight”“[All Fields] AND ““gain”“[All Fields]) OR ““weight gain”“[All Fields])) AND (y_10[Filter])”.

4. (festive season) AND (weight gain),,in the last 10 years”, ((““festive”“[All Fields] OR ““festivities”“[All Fields] OR ““festivity”“[All Fields] OR ““holidays”“[MeSH Terms] OR ““holidays”“[All Fields] OR ““festival”“[All Fields] OR ““festivals”“[All Fields]) AND (““season s”“[All Fields] OR ““seasonability”“[All Fields] OR ““seasonable”“[All Fields] OR ““seasonably”“[All Fields] OR ““seasonal”“[All Fields] OR ““seasonalities”“[All Fields] OR ““seasonality”“[All Fields] OR ““seasonally”“[All Fields] OR ““seasonals”“[All Fields] OR ““seasons”“[MeSH Terms] OR ““seasons”“[All Fields] OR ““season”“[All Fields]) AND (““weight gain”“[MeSH Terms] OR (““weight”“[All Fields] AND ““gain”“[All Fields]) OR ““weight gain”“[All Fields])) AND (y_10[Filter])”.

### 2.2. Study Selection

The search was limited to studies published in English in the last 10 years, with adult participants, but without specifying limits regarding the study design. 

The exclusion criteria were the age of the participants < 18 years, associated comorbidities, articles that addressed intermittent fasting diet or weight loss intervention during the holidays through various diets, and articles that did not address the topic studied in detail. 

We decided to eliminate the studies that include children and adolescents, since both their caloric needs and their food patterns are different from those of adults.

### 2.3. Study Appraisal

The screening process, including title, abstract, and full-text review, was conducted by three independent reviewers (I.M.A., A.M., G.P.) in duplicate. Full texts of selected articles were carefully examined, and papers from the same study were collected. Disagreements that arose during the data abstraction phase were resolved through discussions between the independent reviewers. If a consensus could not be reached, a fourth reviewer (M.M.L.) made the final decision on the disagreements.

## 3. Results

Figure 1 highlights the screening and selection process regarding the articles included in this systematic review. Using the combination of the terms “holiday and obesity”, “holiday and weight gain”, “festive season and obesity” and “festive season and weight gain” we obtained 216 results involving the addressed topic. Thus, after screening, we included only ten articles that addressed the topic of interest.

The characteristics and demographic data for each study are presented in Table 1.

Out of the ten studies, nine were single-sided: USA (6/10), Spain (2/10), UK (1/10), and one took place in multiple countries: USA, Germany, and Japan.

A total of 4627 participants were included in the studies. From ten studies, only seven mentioned the percentage of males/females. Among the seven studies, in six of them, female participants predominate (65–90%), and only one study enrolled mainly men (66–74%).

The mean age of the patients was 40 years in three studies, 55 years in two studies, between 18 and 65 years in three studies, and between 21 and 50, respectively, over 32.2 years in another two different studies.

Regarding the method used, the participants of eight studies had to make at least two visits with the aim of accumulating data. One of them analyzed the change in body weight through wireless scales and another through the self-weighing method.

The participants of five studies were recruited through flyers, advertisements, or emails. For one study, patients from the National Weight Control Registry were approached, and for the rest, conventional methods were used.

## 4. Discussions


**Do we gain weight during holidays?**


Overeating during the Christmas and New Year period is a widespread practice that can be challenging to modify. Various factors contribute to dietary changes during the holidays, including social norms, gatherings with loved ones, and the desire to showcase a favorable material situation. Individuals tend to consume significantly more food in a 24 h period than they would typically, often motivated by the intention to begin the new year with a fresh and healthy lifestyle [20].

Furthermore, festive periods often align with public holidays in many countries, providing ample opportunity for sedentary behavior and overeating. On Christmas days, for instance, individuals may consume up to 6000 calories, which is three times the recommended daily caloric intake. This excessive caloric consumption further contributes to weight gain during this time [15].

In 2016, Elina E. Helander and colleagues, using data provided by wireless scales, obtained information on weight fluctuations of 2924 participants from three countries over a 12-month period (1 August 2012–31 July 2013). The study included 1781 residents of the United States with an average age of 42.2 years and BMI = 27.7 kg/$m^{2}$, of whom 34% were women, and 24% were obese (BMI ≥ 30.0 kg/$m^{2}$), 760 residents of Germany with an average age of 42.9 years and mean BMI = 26.6 kg/$m^{2}$, of whom 34% were women, and 19% were obese, and 383 residents of Japan with an average age of, 41.6 years and mean BMI =24.7 kg/$m^{2}$), of whom 26% were women and 11% were obese.

Participants consistently monitored their weight, and the weight change pattern showed a linear trend until the holiday period. The study specifically focused on the 10 days leading up to Christmas and the 10 days following Christmas. The measurements revealed a weight gain of 0.4% in the United States, 0.6% in Germany, and 0.5% in Japan during this time. Additionally, in Japan, a 0.3% increase in weight was observed during Golden Week, while Germany and the United States saw a 0.2% increase during Easter and Thanksgiving, respectively.

In summary, the study found that participants in all three countries experienced weight fluctuations during the study period, particularly during holiday periods. One concerning observation was that approximately half of the weight gained during these holiday periods was maintained until the end of the study. This suggests that holiday-related weight gain can have long-term effects on individuals’ weight [10].

Another article from 2016 talks about how people can gain weight during holidays and how the number of pounds gained can contribute to annual weight gain. One hundred and twenty-two (122) people were included in this study, and most of them had an above-average body mass index (57—normal weight, 46—overweight, 19—obese). To be included, participants had to be going on a short vacation (7 days to 21 days) to a destination other than their primary residence. The latter were the most prone to weight fluctuations because people with normal weight accumulated 0.28 ± 0.13 kg, while overweight people gained 0.39 ± 0.14 kg, and obese people gained 0.48 ± 0.27 kg. In addition, the stress level before the holiday is high, while during it, the level of activity increases; therefore, the main cause leading to the accumulation of extra kilos would be the caloric intake during this period [11].

Even though in the mentioned study, the weight gain does not seem significant; Dale A. Schoeller analyzed several studies where he observed that overweight or obese people are more prone to gain weight on vacation compared to those of normal weight, which over time can increase the risk of an “obesity epidemic” [21]. He mentions that both Yanovski et al., and Hull et al. reported that subjects who had a BMI ≥ 25 kg/m^2^ were more likely to gain extra pounds. For example, in the study by Hull et al., normal-weight participants gained 0.4 kg while the others gained 0.8 kg [22], and in that of Yanovki et al. at the end of the vacation, normal-weight individuals had a weight gain of 5%, and those with BMI ≥ 25 kg/m^2^ over 11% [9].

Surabhi Bhutani and colleagues followed 23 adults aged 21 to 50 years and BMI 30–39 kg/m^2^ from the Madison metropolitan area, without pathologies or eating disorders, for 16 weeks in a study that aimed to quantify changes in body weight, body composition, energy balance and eating behavior during two consecutive periods: 1. a period of 8 weeks before the holiday (15–30 September–9–25 November) and 2. a period of 8 weeks during the holiday (9–25 November–4–15 January).

The research involved six visits to the study center, three before and three during the holiday period. Initially, anthropometric data were collected, stable isotope-labeled water was administered, urine samples were collected to measure total body water after a night of fasting, and blood samples were collected. In addition, data on participants’ appetite and food preferences were recorded using questionnaires. In week 3, urine samples were collected to assess total energy expenditure, and in week 8, the percentage of adipose tissue was quantified. During the holiday week, the same analysis protocol was used.

In the end, the data obtained in the two periods of the study were compared, and a weight loss of 0.86 kg was observed before the holiday vs. an increase of 0.41 kg during the holiday. Although it was initially thought that the weight change was due to the increase in the energy intake, no significant changes in this direction were observed (+80 kcal/day). However, there seems to be a strong correlation between the change in caloric intake and dining out, a usual habit during the holidays [12].

Another study that investigated the association between increasing BMI and eating out showed that a person does this 1.86 times a week. After analyzing data obtained from participants from smaller metropolitan communities in the Midwest, it was concluded that a positive association exists between BMI change and eating at fast food or restaurants (0.8 kg/m^2^ and 0.6 kg/m^2^) [23]. Even though the study did not show statistically significant fluctuations in weight during the two time periods, there is still a risk that, in the future, the extra caloric intake will contribute to the permanent accumulation of extra pounds [12].

In recent decades, the obesity rate has increased rapidly among young people aged between 18 and 29 years, most of them students. It seems that the first year of college can determine a weight gain of up to 6 kg due to major changes that can increase stress levels. In addition, during the holidays (summer, Easter, Christmas), the social activities include a large quantity of food as a main socialization component, along with excessive consumption of alcohol. A pilot study involving 67 first-year university students at the University of Castilla-La Mancha (10 men, 57 women) with a mean age of 20.4 ± 4.1 years attempted to examine the changes occurring in young people’s lives during the Christmas holidays. Data on initial weight, waist circumference, BMI, and place of residence during the first academic year were collected. During the study, participants weighed themselves four times—December 23, December 30, January 6, and January 13. During Christmas, the body weight tended to increase from an initial mean of 59.6 ± 10.7 kg to 60.2 ± 10.6 kg, but with a decrease to 59.6 ± 9.9 kg in the first week after returning to college. Likewise, BMI showed an upward evolution during the holidays, 21.7 ± 3.1 at the first measurement, 21.9 ± 3.1 at the second, and 21.7 ± 3.9 at the third.

The results, therefore, showed that the students’ weight changed during the holiday, with a similar trend for both men and women. Overall, students with a higher initial weight tended to gain even more weight than those who initially were normal. People who gained weight during this period were able to shed the extra pounds by the end of the follow-up period, but again in the case of those with an increased initial BMI, this decrease was not as evident.

Again, it must be emphasized that the events during this period create an environment prone to the risk of weight gain. Eating high-calorie, high-sugar, high-fat foods is encouraged. The number of calories ingested during a day is not only because the frequency of meals is increased, but also because the amount of food is larger [13]. The social factor is very important, and some studies have shown that sometimes the environment in which lunch is eaten can increase portions by up to 44% [24].


**Traditional christmas foods and calories**


In Romania, the Christmas meal is always plentiful and contains several dishes. Key items include sarmales, periwinkle soup, roast pork, pickles, boeuf salad, house wine, ‘țuică’ and ‘cozonac’ (sweet bread) with walnuts [25].

A British Christmas dinner often includes prawns (shrimp), mince-meat pies, and a roasted turkey as the main centerpiece. In contrast to American traditions of garnishing the turkey with herbs, stuffing, or citrus, the British tradition involves topping the turkey with a bundle of sausages. For dessert, a typical choice is a fruit-packed Christmas pudding, which is a rich and dense steamed or boiled pudding [26].

In Greek households, it is common to celebrate Christmas dinner with roasted lamb as the main dish. However, in the northern regions of Greece, a traditional Christmas food called yiaprakia is also popular. Yiaprakia are brined pork-stuffed cabbage rolls that add a unique flavor and variation to the festive meal. On Christmas Eve, Greeks have a tradition of making Christopsomo, which is a rustic sweetbread. This bread is filled with ingredients such as raisins, nuts, cardamom, and cloves, giving it a rich and aromatic taste. It is often decorated with a cross and becomes a centerpiece on the Christmas Day table [26].

Italy is renowned for its delicious desserts, and during Christmas, different regions have their own specialties. In much of Italy, Christmas Day lunch often ends with panettone, a sweet bread filled with chocolate or raisins. However, in Sicily, the traditional dessert is buccellato, a round cake made with dried figs, almonds, and pine nuts. The distinctive flavor of buccellato comes from the addition of marsala, a fortified wine named after the Sicilian city of Marsala, which is incorporated into the pastry dough before baking. Another famous Christmas tradition in Italy is the Feast of the Seven Fishes, also known as Festa dei Sette Pesci. This dinner consists of a seven-course menu featuring various seafood dishes, such as carp, octopus, clams, mussels, and even fried eel. The final course of this feast is reserved for classic Italian desserts, including panettone or homemade tiramisu [26,27].

In the Catalonia region of Spain, Christmas lunch starts with a traditional dish called sopa de galets. This soup features pasta shells known as galets, which are a beloved specialty in Catalonia. The soup is a labor-intensive dish that requires simmering a broth made from a combination of beef and ham bones, chicken breast, pig’s trotters, and vegetables for several hours. Bite-sized meatballs made from freshly minced beef and pork are then added to the broth along with the galets, creating a delicious and satisfying start to the Christmas meal [28].

On the west coast of Norway, a popular Christmas dish on 24th December is pinnekjøtt. This traditional dish consists of wood-fired lamb ribs that are first dried, cured, or smoked before being slow-cooked over birch wood. The result is juicy and tender meat with a delightful smoky flavor. Pinnekjøtt is typically served with traditional accompaniments such as swede and carrot mash, which adds a touch of sweetness to complement the rich flavors of the lamb. Lingonberry jam, a sweet and tart condiment, is often served alongside to balance the savory taste of the meat. To complete the meal, a shot of akevitt, a Scandinavian spirit infused with fennel, caraway, and star anise, is often enjoyed as a traditional pairing with pinnekjøtt [28].

In Germany, roasted duck, goose, or rabbit are popular choices for the main course of Christmas dinner. These meats are often accompanied by delicious side dishes such as sausage stuffing, potato dumplings, and red cabbage, which add a savory and hearty element to the meal. Feuerzangenbowle, which translates to “fire tong punch”, is a unique and extraordinary version of mulled wine. This special drink is commonly served in German Christmas markets throughout December. It begins as a typical glühwein, which is warm red wine infused with flavors such as orange peel, cinnamon, and cardamom [26,28].

Foie gras is indeed a delicacy that is closely associated with Christmas in France. It is a rich and creamy dish made from the liver of a specially fattened duck or goose. During the holiday season, foie gras is often enjoyed in various ways.

One popular way to savor foie gras is by spreading it on different types of bread toasts. Whether it is a simple baguette slice or a more decadent brioche bread or gingerbread, the buttery texture and unique flavor of foie gras pair well with the crispy or soft bread. Baked turkey is also a popular and classic Christmas dish. Side dishes include green beans wrapped with bacon, truffle mashed potatoes. A large cheese platter with a variety of different kinds of cheese is a must-have during Christmas celebrations. Bûche de Noël, also known as the Yule log, is a classic dessert that holds a special place in French culinary traditions during the holiday season. The dessert is typically made by rolling a sponge cake into a cylindrical shape, often filled with flavored creams or mousses. The exterior is coated with buttercream or ganache, resembling the bark of a tree. The log is then decorated with festive touches, such as powdered sugar to resemble snow, chocolate shavings, or tiny figurines depicting Christmas scenes [29].

The traditional Christmas dinner in America has its roots in British cuisine. It typically includes roasted root vegetables as a side dish, mashed potatoes, gravy, and a stuffed roasted bird as the centerpiece. In the Southern region, which has a significant population with British ancestry dating back centuries, Christmas is celebrated with various variations of country ham or Christmas ham. Moreover, different regions across America offer diverse regional meals during Christmas. For example, in Virginia, there are oysters, ham pie, and fluffy biscuits, which pay homage to the English founders who settled there in the 17th century [30].

Japan, although not historically a Christian country, has a predominantly atheist population. As a result, some Christian events are celebrated in Japan mainly for commercial reasons. Due to the difficulty in obtaining turkey, chicken has become a popular choice for Christmas day meals. In the 1970s, KFC began offering Christmas chicken deals, which further popularized the tradition of eating chicken during the holiday season. Various chicken dishes such as kara-age, teriyaki chicken, or fried chicken are commonly prepared for Christmas. Additionally, potato salad is a common side dish, possibly influenced by German culture as about 40% of Germans eat potato salad on Christmas Eve. Another popular dish served during this time of the year is cream stew, a Yōshoku dish consisting of meat (usually chicken or pork), mixed vegetables, onion, carrot, potato, and cabbage cooked in a thick white roux. Cream stew is considered a winter dish in Japan as it provides warmth during the cold season. It is believed to be a Western meal and is served on Christmas due to its origins in Western countries. After the main meal, it is customary to serve a layer cake topped with strawberries. The tradition of Christmas cakes in Japan dates back to 1910 when a cake shop called Fujiya started selling decorative Christmas cakes, which became popular among the Japanese population [31].

In Colombia, the month of December is dedicated to celebrations, and the Christmas season officially begins on December 7th, known as the Day of the Small Candles (Día de las Velitas). Since the main Christmas meal is typically eaten close to midnight, it is common to serve snacks throughout the 24th. Fried cheese balls (buñuelos), cheese puffs (pandebonos), tamales (a dish made of meat, vegetables, and corn puree steamed in banana leaves), and natilla (a custard-like dessert) are popular options. Tamales are also commonly served for breakfast. For Christmas dinner, Colombians often enjoy a pork roast or a dish made with pork, as beef is usually consumed more frequently throughout the year. If the family gathering is large enough, a lechona may be served. This is a whole pig that is roasted for hours and stuffed with vegetables and rice. Ajiaco, a chicken and potato soup from the capital city of Bogotá, can be served as an alternative.

Side dishes typically include potato salad (made with potatoes, peas, and mayonnaise), boiled potatoes, and rice. Desserts often consist of natilla (the milk custard), manjar blanco (a thick caramel cream made from sugar and milk), or torta negra (a dense fruit cake similar to German stollen, prepared well in advance to develop its flavor) [32].

All of the above dishes and their nutritional information can be observed in Table 2.


**Behavioral patterns**


The academic holidays, particularly Christmas, are periods in which weight gain is often more pronounced. During Christmas, individuals tend to consume larger quantities of food and drinks compared to the duration of the holiday period. In Spain, the Christmas holiday typically lasts for 2 weeks, which is shorter compared to the summer break that lasts between 8 and 10 weeks. However, during this shorter holiday period, people often have frequent gatherings and reunions with family and friends, where food plays a central role in the celebrations. This can contribute to increased consumption and potential weight gain during the Christmas season [82].

Some very interesting articles investigate the underlying behavioral patterns that may lead to overindulgence that provide overall and context-specific explanations of people’s behavior regarding food choices during holidays.

The way we perceive food involves multiple senses, as our experiences are shaped by the interplay of different senses throughout the process of consumption [83]. It is not just the taste or aroma of the food itself that affects our enjoyment of a meal (intrinsic factors), but also various contextual elements such as the arrangement of the table, the background music, the ambiance of the dining area, and even the specific scents associated with the location (extrinsic factors) [84].

A good example is an early study that has identified an external cue that plays a role in guiding the food intake of obese individuals, as compared to those with normal weight [85]. This cue is related to the smell and palatability of food rather than their internal bodily states. In 2010, Ferriday and Brunstrom conducted a study that demonstrated how a brief exposure to the sight and smell of pizza resulted in increased salivation and a stronger desire to consume pizza and other foods in overweight participants, in comparison to those with normal weight [86].

In a 2023 study conducted by Tran et al., the researchers examined how the multisensory aspects of eating experiences are influenced by seasonal changes in two countries: Colombia and Norway. Interestingly, the results showed that Colombian participants associated salty and fried foods with the cold season, considering them to be energy-dense options. On the other hand, they associated sweet beverages, such as soda, with the hotter months. These findings were somewhat consistent with the associations made by participants from Norway, who also associated colder months with salty and fattier food [87].

In 2016, Suzanne Higgs and Jason Thomas draw attention about other important aspect: social context that influences the amount of food intake [88]. They cite a series of studies regarding the impact of social influence on eating. Thus, some reveal that we are likely to model the behavior of the person we are eating with. Therefore, if they are eating a large amount, we are more likely to consume more than we would have eaten alone. Other studies, based on food diaries, observation, and experimental studies, revealed that we are more likely to eat more if we are in a group compared with eating alone.

Christmas dinner is all about reuniting the family, food celebration, and the way the house, room, and table look, smells, and attracts everyone around it. Considering that appearance, taste, odor, texture, temperature, and flavor, along with taste, all impact the amount of food intake, Wadhera Devina and Calapdi-Phillips Elizabeth propose a review that identifies various visual factors associated with food, such as proximity, visibility, color, variety, portion, size height, shape number, volume and the surface area that affect foot acceptance and consumption [89]. The authors used the combination of dietary intake and visual cue keywords to generate scientific, peer-reviewed articles from prestigious databases, such as Medline, Nutrition, PsychINFO, and marketing databases and included both children and adults.

The review depicts the results from many articles clustered on key themes that affect food intake. Thus, regarding proximity and visibility, if the food looks attractive, then increasing visibility will result in greater energy intake compared with one that is unappealing. Another article deals with how much food has already been eaten, which affects food intake. Thus, the amount of chicken wings or pistachios consumed is higher if the empty bones or pistachio empty shells were removed from the subjects’ sight. Another report shows that increasing proximity to food can also increase food intake because it is more visible to consumers. These findings impact the amount of food consumed on Christmas meals, as it is on the table very close to family and guests.

The variety of types of food on the Christmas meals in appearance, texture, taste, and flavor also affect intake, according to the research cited above, by changing perceived, quantity estimators.

Portion size is another key contributor to energy intake and body mass index (BMI). According to the authors, people eat more if larger food portions are served rather than smaller portions. People also take large bites when served larger portions of food. This leads to gorging. This does not allow sufficient time to release regulatory peptides required to develop satiety. If you do not feel full, you will continue to eat, increasing thus the food intake.

Another factor that influences judgment of the amount of food consumed is the area occupied on the plate. If served on larger plates or bowls, food may be displaced away from the edge of the plate, resulting in a significant underestimation of food amount (contrast effect). However, if served on smaller plates, a substantial overestimation of food (assimilation) will occur. These effects arise due to an optical illusion called the Doelboeuf illusion, as cited in in the review.

A study published by Laurier and Wiggins provides an extended review of the interactions during the family meal focusing on completion and expression of satiety. It is based on qualitative research that uses 90 h of audio tapes, family meal time conversations, and an additional eight hours of videotaped family meal times. While focusing on children of different ages (preschool, primary school, and teenagers) on three occasions (a barbecue routine meal and Christmas), some of the behaviors and routines described are very well suited for understanding overeating and obesity. The family transmits and transforms many eating practices to children, including whether and how to comply (or not) with the request to finish. Some of these learned things will stay with the child during adulthood. The plea to finish described in the article is a series of requests of decreasing amounts from ‘little piece’ to ‘very, very little’ and to ‘crumbs’. This, in turn, is replicated by children when requesting the ‘reward’ of the marzipan of the Christmas cake. In adults, during Christmas meals, sometimes filing to finish the portion may lead to being interrogated, persuaded, and worried about (whether you may not like the food) [90].

Social motivation, such as risk-taking and conformity, affect food intake. Kimura et al. explored whether partner presence (pair vs. individual) would affect unfamiliar food intake based on a series of experiments [91]. The study included students from Tokyo, Denki University, with a total of 19 pairs (38 participants, eight females, and 30 males) and 19 individuals (five females and 14 males) with an average age of 20 years. The study did not control the gender combination within pairs because it focused on all pairs consisting of friends, not acquaintances or strangers. The research method measured the amount of participant food consumption and subjective evaluation of food. It was based on an experiment that had a 2 (food familiarity, unfamiliarity vs. neutral within subjects) × 2 (partner presence, pairvs. individual: between subjects) mixed design. It was interesting to find that the ratio of participants who consumed all three kinds of unfamiliar foods was higher in the paired condition compared with the individual situation. The results also showed that in pairs, the participants tried unfamiliar snacks, even after the pair expressed a negative evaluation of the snack. Such results prove that social motivation and conformity affect unfamiliar food intake in co-eating situations. We also believe, as the authors highlighted, that the extension of time spent at the meal by the presence of a companion is one determining factor of social facilitation in the amount of food consumed.

This research is essential for a better understanding of food intake during Christmas, as the Christmas meal would contain at least some unfamiliar dishes (rarely prepared and served outside Christmas dinner). The results described support the behavioral patterns of increased food intake when you have companions and spend a longer time around the table during the meal.

Food consumption patterns exhibit seasonal variation, especially in countries with significant seasonal changes in climate. An analysis of factors affecting seasonal differences in food consumption has been addressed in an extensive review. Charles Spence revealed in his publication that cultural and ritual factors combined with ever-increasing sophisticated data and driven marketing could prove to be more important than nutritional, environmental, or psychological factors when explaining why we eat different food in different seasons. From the first lines of the article, he builds the explanations that consider the cultural/ritual associations, especially when we analyze food rituals that have been shown to enhance consumption. Beyond these, the climate has a very important role to play. This is due to the environmental modifications that imply changes in ambient temperature and humidity, perceptual and psychological factors, especially in mood change during the cold winter with little light. Of equal importance may be the period of fasting occurring in some cultures before Christmas, which might instill in some people the desire to have a healthy start at the beginning of the year. Nowadays, with the impact of globalization, it is much simpler to buy, especially in Western civilization, almost any kind of food all year round.

Therefore, the climate/psychological factors, which, some time ago, played an essential role in determining what we ate and when we ate it, have a diminishing effect [92]. This increases the impact of culture/ritual and human psychology due to the importance of food culture and ritualized eating occasions such as Christmas and other ritualistic religious celebrations. The size and significance of behavioral change in the population around Christmas provide marketers with a great opportunity to promote specific foods that may differ from the typical fayre that is consumed during the rest of the year. Some articles cited in the review [92] report a decrease in the sale of ready-to-eat cereal by 10–20% and an increase between 50 and 225% of piecrust in the week before and during Christmas holidays.

Several research results provide strong evidence that people adjust their food intake, aligning with perceived eating norms, in order to convey positive impressions of themselves to other people and that they eat more when eating with friends and family, compared to when eating alone [93]. Through social facilitation, some individuals may eat as much as the person in the group who is eating the most in order to maximize their intake of palatable food. Another possibility is that some people may overestimate the amount eaten by other people around the table, and they increase the amount of food to match the perceived intake of others.

Reviewing the research literature, the paper explores “social facilitation of eating” defined as the augmentation of food intake in the presence of co-eaters, reported in a series of food diary studies conducted by John de Castro. Such social facilitation is essential because it proves that the tendency to eat larger meals is not restricted to particular social occasions (such as family gatherings, events, or celebrations, such as Christmas, etc.). Evidence of social facilitation of eating was reported for meals consumed during weekdays and weekends, across all types of meals (breakfast, lunch, dinner), and for meals eaten with and without alcohol.

Evidence shows that people do not fully compensate for additional intake, and therefore, eating socially will increase calorie intake and promote weight gain [93]. So, people need to acknowledge the impact of social influence on food intake. Participants watched a video of someone else or themselves eating with another person. They successfully recognized (through mimicking) the role of social influence when they watched someone else eating but did not when they watched themselves. While further research may be required, the consequences for Christmas dinner and meetings are important because some people would need to be made aware of the increased food intake due to social eating.

The research of Helen K. Ruddock et al. also highlights some interesting results related to mood and appetite processes. Meals are more enjoyable when eaten socially, and therefore, people will consume larger portions. This was proven through the idea that a positive mood is more likely to be reported during a social eating occasion. Christmas meal comes with a positive food, and therefore, people are likely to indulge in higher amounts of food intake. The appetite process is affected by social leading through distraction. Research has shown that eating while engaging in distraction, such as engaging in a social environment or watching television, attain weight, feeling of fullness, and sensory of specific satiety, causes people to have larger intakes of food compared to eating without distraction [93].


**Moderation or seasonal dieting?**


To prevent weight gain, even in small amounts, it is important to practice moderation when consuming snacks and calorie-dense beverages such as eggnog. Additionally, increasing physical activity levels, such as going for brisk walks after meals or dancing on New Year’s Eve, can help stabilize blood sugar levels and burn extra calories. A recent study has even suggested that low-volume, high-intensity interval training can effectively reduce hyperglycemia in individuals with type 2 diabetes [94].

The term “seasonal dieting” refers to a pattern of changes in dieting behaviors that typically start or become more intense during the spring season. This is often driven by individuals anticipating heightened body dissatisfaction as summer approaches. People may feel pressure to achieve a certain body shape or size in order to conform to societal ideals associated with the summer season, such as wearing swimsuits or revealing clothing. As a result, they may engage in dieting or restrictive eating practices to try and attain their desired appearance [95].

In a study conducted by Park et al., searches for the term “diet” on Google and Naver (a popular search engine in South Korea) were analyzed in different countries in the Northern Hemisphere and Southern Hemisphere from 2004 to 2018. The findings revealed that searches for “diet” were at their lowest in December and January. The authors interpreted this pattern as being influenced by the New Year period, during which people typically make resolutions, often focusing on weight loss, following the Christmas holidays [96].


**Deeper understanding of psychological reasons**



*Desire to indulge*


One’s pursuit of sensorial or experiential enjoyment is often labeled as “*hedonic consumption*”. Hu and Min highlight the importance of not mis-defining or mis-operationalizing the term “indulgent consumption” as simply being hedonic (such as having a snack, eating ice cream and cookies, or enjoying a dessert). The authors adopted a definition proposed by an author they cited, which defines “*indulgent consumption*” as “time-inconsistent preferences, or a tendency to overweight short term’s rewards relative to more distant ones and the tendency in the short term to ignore the cost of one’s actions”.

The fundamental difference between the two concepts is due to the goal conflict. In order to examine its impact on individuals’ subsequent cognitive and affective processes, the authors designed a scenario-based quasi-experiment with 2 (goal conflict yes vs. no) × 2 (justification for consumption yes vs. no) groups. Goal conflict has been manipulated based on the health status of a fictitious character called Sam, who had health issues and needed to control intake, or he was perfectly healthy and not on a diet. Consumption manipulation was based on consumption occasions (regular dinner vs. birthday celebration). The participants, who were randomly assigned to one of the four conditions, had to respond to questions regarding their perception of the consumption, such as perceived enjoyment indulgence and self-control failure, after they were given a scenario to read.

The study revealed that hedonic consumption is perceived as self-control failure only when goal conflict exists and, therefore, recognized as less enjoyable and more indulgent [97].

While the research [97] addressed the hospitality industry, its findings may be relevant for people eating during the Christmas holidays. Thus, if family and guests recognize a goal conflict, a legitimate reason for gratification (an extra portion of the delicious but high-calorie food on the table) over long-term goal pursuing effort (losing weight), they may enjoy the hedonic consumption more than those without goal conflict.

Another interesting study on altruistic indulgence explored how and why people in certain social contexts are altruistically motivated to consume high-calorie foods to make other people feel comfortable and pleasant, especially for those with whom they are friends and family [98].

Altruistic indulgence refers to the voluntary consumption of high-calorie foods by someone with the altruistic motive of making other people feel comfortable and pleasant. It manifests as a “healthy-causes-guilt” context [98]. It depicts all situations in which, before making a food choice, someone is aware that their choice of a healthy food will elicit negative feelings of guilt from another person.

The paper is based on a field study that included 649 transaction receipts from a coffee shop during one week. There were some solo, and the rest were dyad purchases. There were four types of purchases: (1) solo, (2) first-orders of dyad purchases, (3) second-orders of dyad purchases that started with a low-calorie diet, and (4) the healthy-causes-guilt context where second-order purchases started with a high-calorie choice [98].

Whilst the study took place in a coffee shop and revealed that altruistic indulgence was more likely to occur with a friend, such behavior may also occur during the Christmas table where some family members (as an indulgent companion—the one that makes first the choice of unhealthy food) may influence the other to also choose higher amount/unhealthy food.

Through awareness of such a phenomenon, people around the table may understand the mechanism and make better decisions for themselves.


*Stress associated with celebration*


In order to analyze stress related, eating, the research analyzed 158 subjects that self-completed daily records of stress in eating for 84 days [99]. As a response to daily stressful problems, the subjects were more likely to eat less than usual than to eat more than usual. The results revealed a substantial increase in the likelihood of eating, less as the severity of stress increased, while the likelihood of eating more did not change with an increase in the severity of stress. Many individuals use larger amounts of food as a source of comfort in the face of stress [99], therefore programs for obesity and eating disorders include relaxation and stress management approaches. Other researchers reported that stress increases eating in some cases and decreases eating in others. Such behavior could be due to environmental factors that may lead some humans to eat less when stressed. If an individual’s primary means of coping with stress is eating, that would lead to an increase in intake during stress. However, those individuals who use other means of coping may have limited attention to devote to eating and, therefore, may eat less [99]. These results are supported by a study that analyzed smoking, stress eating, and body weight [100].


*Emotional eating*


Researchers defined emotional eating (EE), a state that is often encountered during the winter holidays as “*the tendency to overeat in response to negative emotions, such as anxiety or irritability*”. Highly palatable foods are those with low nutritional value, but high in fat, sugar and salt. Empirical evidence indicates that by suppressing the hypothalamic-pituitary-adrenal axis response, highly palatable foods protect against stress [101].

Research also revealed that loneliness is associated with preferences for palatable over healthy food [102].

A persons’ behavior will depend to a great extent on two key variables: the situation (or context), and the person (their own characteristics and available resources).

There are three important coping strategies with stress: avoidance, emotion-oriented and task-oriented coping strategies. Engaging in a substitute task (also known as avoidant distraction) or via social diversion, represents avoidance coping strategies. Elevating negative emotions associated with a stressor represents emotion-oriented coping, and addressing and dealing with a stressor are the task oriented coping strategies. Music was reported as a commonly used strategy for coping of people who engage in EE [101] which has important implications for advising people who engage in EE to listen to music for discharge. Such strategies may also be used during Christmas.

Sometimes parents feed their children in order to relieve the child’s emotional distress, which is also called emotional feeding which may be considered as part of EE. This may cause recurring emotional eating is the child has been taught to use food as a coping mechanism [103]. Another important coping mechanism with EE is mindfulness. This is an awareness that emerges through deliberately paying attention to the present moment without judgment. Mindful eating practices may decrease emotional eating behaviors through interventions on parent, mindful eating and parent mindfulness. Being aware of the power of mindfulness, such strategy for coping could be taught and used during Christmas.

A study on eating pathology in infertile woman revealed that both perceived, stress and avoidant coping styles are important factors that influence the eating pathology. There were six avoiding coping styles used in the research: self-blame, disengagement, venting, substance abuse, self-distraction, and denial that mediated the relationship between perceived stress and eating pathology. Restrained, eating concern, weight concern, shape concern, and disordered eating behaviors were the eating pathologies used in the research [104].

Increased eating associated with stressors also represents an important method of coping, even if it has negative consequences [105]. The research shows that EE as a coping mechanism may be triggered by unsupportive social interactions, especially if one considers that eating might actually be a way of coping with adverse events as well as the propensity to adopt a coping style that does not requires reliance on others, such as emotional or avoidant coping. Christmas family meetings may sometimes provide an environment favorable for the occurrence of unsupportive social interactions among some of the friends and family members and thus to induce EE as a coping mechanism.


**Nutritional patterns during COVID19 pandemics**


The worldwide implementation of strict non-therapeutic measures during the lockdown had a notable effect on not just people’s mental well-being, but also their personal dietary habits and lifestyle. This included changes in access to and availability of food, patterns of food consumption, as well as the type and amount of physical activity undertaken including the cold season [106,107].

In the absence of effective medications for COVID-19, incorporating nutritional dietary supplements that contain essential elements and vitamins may be the most effective way to boost immunity in adults. However, the lockdown imposed during the pandemic has had significant repercussions on various health aspects. These include irregular or unhealthy eating habits, lack of physical exercise, and substance use, all of which can increase the risk of contracting the disease and contribute to feelings of fear, anxiety, depression, and other mental disorders [108]. The emotional strain associated with COVID-19, such as anxiety and depression, can also lead to excessive consumption of carbohydrate-rich foods through “food cravings” or “emotional eating” [109]. Additionally, the prolonged confinement at home can create a sense of monotony, leading to overeating as a means to alleviate boredom or dullness [110].

In addition to the existing global health concerns of lower physical activity levels and increased obesity, the COVID-19 pandemic has further highlighted the need to address these issues. Uncontrolled unhealthy food consumption and limited physical activity can have a significant impact on overall health and increase the risk of various diseases such as obesity, diabetes, cancer, and cardiovascular conditions, as observed in other developing countries [111,112].

Due to the disruption of work routines caused by quarantine, individuals may experience boredom, which has been also linked to increased consumption of fats, carbohydrates, and proteins [113]. Additionally, the constant exposure to pandemic-related information without respite during quarantine can lead to stress, prompting individuals to seek solace in sugary “comfort foods” and potentially overeat. This specific desire for certain types of food is referred to as “food craving”, which encompasses emotional (intense desire to eat), behavioral (seeking food), cognitive (thoughts about food), and physiological (salivation) processes. Interestingly, there is a gender disparity in food craving, with higher prevalence observed in women compared to men [114].

During the pandemic, there has been a greater focus on home-based exercises to maintain physical health, as movement outside the home has been strictly discouraged. Additionally, it has been predicted that the COVID-19 pandemic could disrupt the availability and accessibility of food due to restrictions placed on transportation systems, potentially resulting in increased food insecurity [115].


**Prevention strategies for weight gain**


It is common to say, “I’m too occupied with Christmas preparations to find time for exercise”, or “I receive numerous invitations to indulgent Christmas parties”. “While we may enjoy the holiday season, we are allowed to overindulge in food and drink”. These are just a few examples of the excuses some use to justify their behavior during the winter holidays, even for those who maintain a regular exercise routine throughout the rest of the year. The permissiveness of this period when it comes to various foods and the consumption of alcohol (also rich in calories), does not completely exclude a minimum of measures that can be taken easily.


**Self-monitoring**


An excellent strategy to prevent the accumulation of extra pounds during the holidays would be to implement small lifestyle changes, even if attention is directed to other aspects of life during this period, but also to be aware of the effects of an unbalanced diet even on a short period of time.

From the National Weight Control Registry, 683 people were recruited [24]. They had an average age of 54.6 years and BMI of 26.9 kg/m^2^, lost at least 13.6 kg (30 pounds) at one point, and maintained their weight for at least a year afterward. Participants were given a list of 18 weight management strategies from which to choose a top three, based on their goals for the festive period. To quantify the effectiveness of the preferences, the participants were measured, weighed, and questioned about the self-imposed goals for the next period. Seven (7)% had no weight loss goals, 47% wanted to maintain their weight, 11% hoped to limit the accumulation of extra kilograms, 35% had a weight loss plan, and 0.3% did not have any objective. In addition, for the pre-holiday assessment, participants placed themselves either in the “I plan to use it” category or in the “I don’t plan to use it” category.

After the holiday, each strategy was placed into a new category based on how often it was applied. The strategy with the highest applicability rate (100%) was choosing healthy snacks while traveling, although a small number of people planned to approach this strategy. Focusing on other aspects of the festive period and not on food ranked second in the list of participants’ choices (95.8%). In contrast, daily weighing (90.5%) and healthy food choices (93.7%) were the strategies associated with weight changes. This is not surprising, as a high level of attention to meals helps manage caloric intake simply by prioritizing healthy foods over those considered unhealthy.

It is important to note the fact that all choices were made according to the participants’ preferences in relation to the goal they wanted to achieve. Thus, those who aimed to maintain/lose weight approached 12/18 strategies, those who tried to limit the accumulation of extra kilograms 11/18, and those who had no objective nine/18 strategies. So, it seems that the prevention of obesity during this period depends not only on the number of strategies we have at hand and can easily approach, but also on the degree of involvement of people in this endeavor [14].

Another study, the Winter Weight Watch Study, looked at the influence of minor lifestyle changes among 272 adults over 18 with a BMI ≥ 20 kg/m^2^. The participants were divided into two equal groups: one that received a leaflet with information on how to approach a healthy lifestyle but no diet advice, and one in which members received ten tips on maintaining their body weight, and were encouraged to weigh themselves regularly and given information about the type of exercise they could do to burn off the extra calories consumed at the festive meals. Baseline assessments took place in November and December, with follow-up assessments in January and February (4–8 weeks post-baseline). The aim of this study was that during the festive periods, the participants should not gain more than 0.5 kg. At the end of the study, it was observed that in the intervention group, the mean unadjusted change in weight was −0.13 kg, and in the control group, which only received the leaflets, it was +0.37 kg.

By monitoring their weight regularly, participants reflected much more on the food and drink they consumed. Individuals in the intervention group were more restrained than those in the control group in consuming high-calorie foods and alcoholic beverages. To the question “Have you consumed alcohol in the last week?” positive responses were found in 70% of the people in the control group, compared to 61% among those in the intervention group. Therefore, it seems that small changes in this period could help avoid the accumulation of extra pounds and thus prevent obesity [15].

To test the effectiveness of daily self-weighing (DSW) using a visual graph as feedback, Sepideh Kaviani et al. recruited 111 participants older than 18 years and with BMI ≥ 18.5 kg/m^2^ online or using advertisements displayed in restaurants or shopping centers. As in the anterior study, participants were equally divided into a control group and a target group. The target group was instructed to weigh themselves daily and not exceed their average weight since the beginning of the study period, but without receiving instructions on how to reach this target. The control group did not receive this kind of training. In addition, all participants underwent anthropometric measurements, biological samples were taken for lipid metabolism analysis, and their blood pressure was measured before and after the study. There were three evaluations during the study, one before the holidays (after Thanksgiving), the second after New Year’s Day, and the third 14 weeks after the second evaluation.

The awareness of weight and possible fluctuations had, it seems, a particular impact, so that participants who weighed themselves had better control over the frequency of meals and the amounts served. Moreover, in the target group, there was a decrease of −1.46 ± 0.62 kg among overweight/obese people and a maintenance of weight in normal weight of 0.33 ± 0.27 kg, opposite to the control group. Likely, participants with a BMI above the normal were much more influenced by their weight during the experiment, which could help them over time and prevent the accumulation of extra pounds during the festive period. In the control group, a substantial increase in weight of +2.65 ± 0.33 kg was observed, but most of the accumulated kilograms were lost by the end of the study (57%). Interestingly, men lost 95% of the total accumulated up to the last visit, compared to women, who managed to lose 77% of the total.

In conclusion, even if self-monitoring does not seem to be such an important tool, it can eventually become an addition to the strategy of maintaining an ideal weight [16].


**Prevention at work**


Jobs offer a unique opportunity to promote a healthy lifestyle. People working in the same place interact a lot with each other and create close relationships; as a result, some policies can be much easier to implement than in the community.

Diet and lifestyle improvement programs are often recommended when employees are already obese, instead of emphasizing the importance of prevention. As previously mentioned, the holiday season is problematic for weight. Accordingly, Mark G Wilson et al. conducted a pilot study that included 239 state workers, with an average age of 47.1 years, from the same department. The test was divided into two phases, each running for 10 weeks, from the end of October to mid-January (2015–2016). In the first stage, 100 employees participated; in the second, 139 employees, of whom 36 were also in the first stage.

The study was based on teamwork, self-monitoring, regular weighing, and implementing positive habits. Employees met with a kinesiology staff member and were given a list of activities designed to promote healthy eating habits and physical activity. In order to achieve the goal of compensating for the increased caloric intake and reduced physical activity, the participants were proposed a caloric deficit of 500 kcal/day.

After the initial weigh-in, teams were formed, and each group received points throughout the study for participating in the weekly activities and for completing the bi-weekly weigh-ins. At the end of the 10 weeks, awards were given to celebrate achievements.

Participants were weighed at the beginning and end of the study, measured every two weeks, and participated in a pre-and post-study survey related to dietary habits and physical activity.

Weight loss was significant throughout the program (from a mean weight of 196.7 lb/89.2 kg to 192.3 lb/87.2 kg), increased time spent on physical activity, reduced fast food consumption, and implementing more fruit and vegetables into the diet.

To examine weight maintenance between programs, the data of the 36 participants who agreed to repeat the test were analyzed. On average, they lost 6.9 lb/3.1 kg in the first year, gained 8.7 lb/4.0 kg between programs, and lost 4.3 lb/2.0 kg in the second year, resulting in a total loss of 2.5 lb/1.1 kg. It appears that of 31% who maintained or lost weight, 44% gained less than 10%, and 25% of them gained 10% or more throughout this period between programs.

Therefore, teamwork that aims for a healthy lifestyle, including at work, can be an excellent way to prevent the accumulation of extra pounds during the festive periods of the year [17].


**Physical activity**


Even though it may feel challenging to exercise during the holiday season, incorporating brief moments of physical activity throughout the day can contribute to maintaining the physical and mental well-being. It is important to remember that fitness during this time should not be a source of stress or obligation. Taking rest days and allowing time off to spend with loved ones is just as important.

In 2013, Stevenson et al. conducted a study that included 148 participants (100 women, and 48 men aged 18–65 years old, with an average BMI of 25.1 ± 0.5 kg/m^2^) in order to evaluate whether exercises performed regularly during Christmas have an effect on body weight. The initial assessment was performed in mid-November, and the final evaluation was in early January. Characteristics analyzed were weight, body fat percentage, blood pressure, and self-reported physical exercise. Depending on the time devoted to physical exercise throughout the study, the participants were divided into two groups: “athletes”, who did physical exercise of moderate intensity > 150 min per week, and “non-athletes” who did physical exercise more less than 150 min per week.

During the entire festive period, the participants gained 0.78 ± 0.1 kg, and their fat percentage increased by 0.5 ± 0.2%, without significant differences between non-athletes and athletes (BW: 0.86 ± 0.2 kg vs. 0.70 ± 0.1 kg; BF%: 0.7 ± 0.3% vs. 0.3 ± 0.3%; BMI: 0.3 ± 0.1 kg/m^2^ vs. 0.2 + 0.1 kg/m^2^). A considerable difference was observed in the case of systolic blood pressure (SBP), as non-athletes showed an increase of +4.8 ± 1.6 mmHg, compared to athletes +0.3 ± 1.6 mmHg.

It should be noted that obese subjects showed a greater increase in fat percentage compared to normal weight and overweight subjects: 1.6 ± 0.5% vs. 0.2 ± 0.2% vs. 0.5 ± 0.4%. Moreover, the SBP value of the obese participants increased by 6.6 ± 2.3 mmHg, compared to 0.7 ± 1.6 mmHg in the normal weight group.

Therefore, more than lack of exercise, initial body weight influenced changes over the holiday period, with exercise not having a protective role against accumulating extra pounds, especially among people with a BMI above the normal limit. However, regular physical activity may have a protective role against blood pressure changes, as there was a trend towards increased SBP in those who were sedentary throughout the examination [18].

On the other hand, in a randomized controlled study, an attempt was made to evaluate the effect of HIIT physical exercises on a cycle ergometer during the three weeks that include the winter holidays (December 20–January 10) on 38 men with an average age of 57 ± 8 years, BMI 32 ± 5 kg/m^2^, that had diagnostic criteria for metabolic syndrome.

To be classified as metabolic syndrome, participants had to present 3/5 risk factors: increased abdominal circumference, high blood pressure, elevated fasting blood glucose, hypertriglyceridemia, and low high-density lipoprotein values.

Initially, all participants performed 12 weeks of high-intensity training on a bicycle, a week before Christmas were evaluated, and the first part of the experiment ended. Later they were divided into two groups—22 people who continued to train (TRAIN) and 23 people who stopped training (HOLID), but all were encouraged to maintain their eating habits and lifestyle.

The workouts lasted 43 min, which included a 10 min warm-up (70% of maximum HR), four repetitions of 4 min of exercises in which 90% of maximum HR was reached, 3 min of recovery with 70% of maximum HR, and 5 min of recovery three times a week for three weeks.

Weight, height, abdominal circumference, and body composition by bioimpedance were assessed by the same researcher at the start and end of the program. The participants had their systolic, diastolic, and mean blood pressure values measured after 15 min of rest. A blood sample was taken to analyze blood glucose, insulin, and lipid profile (HDL-cholesterol, triglycerides, total cholesterol).

At the end of the study, the participants were questioned about the calories consumed and physical activity during the festive period. No differences were observed between the two groups. On average, the subjects consumed 2596 ± 93 kcal/day in the last three days of the experiment and estimated a physical activity of 1810 ± 1524 MET/min/week.

When comparing before and after the holiday, a significant increase in mean blood pressure was observed in the HOLID group 94.0 ± 0.6 mmHg vs. 97.1 ± 8.9 mmHg, compared to those in the TRAIN group, where no significant changes were recorded at 97.7 ± 7.9 mmHg vs. 95.3 ± 8.2 mmHg. In the sedentary ones, the abdominal circumference increased from 108.1 ± 10.3 to 110.1 ± 9.4 cm. Mean weight increased from 91.3 ± 13.0 kg to 92.0 ± 13.2 kg in the HOLID group, while in the TRAIN group, it decreased from 99.2 ± 19.6 to 98.98 kg. Body weight change was correlated with increased LDL-cholesterol, SBP, and worsening lipid oxidation capacities during exercise. Reduced lipid oxidation capacities were also correlated with increased SBP, and high triglycerides with decreased insulin resistance.

In conclusion, a supervised exercise program over the Christmas period is an effective strategy for avoiding weight gain and its deleterious effects in people with metabolic syndrome or weight problems [19].

Even if the effectiveness of sports cannot be demonstrated for a short period of time such as the winter holidays research demonstrates that participating in regular moderate-to-vigorous physical activity provides many health benefits. Some benefits of physical activity can be achieved immediately, such as reduced feelings of anxiety, reduced blood pressure, and improvements in sleep, some aspects of cognitive function, and insulin sensitivity. Other benefits, such as increased cardiorespiratory fitness, increased muscular strength, decreases in depressive symptoms, and sustained reduction in blood pressure, require a few weeks or months of participation in physical activity. Physical activity can also slow or delay the progression of chronic diseases, such as hypertension and type 2 diabetes. Benefits persist with continued physical activity.

It has been estimated that people who are physically active for approximately 150 min a week have a 33 percent lower risk of all-cause mortality than those who are not physically active. Physical activity and caloric intake both must be considered when trying to control body weight. Because of its role in energy balance, physical activity is a critical factor in determining whether a person can maintain a healthy body weight, lose excess body weight, or maintain successful weight loss [116].


**Geriatric population**


Eating problems can be frequently observed in the elderly. Researchers include chewing and swallowing complications, xerostomia, poor appetite or oral health related eating problems as some of the most prevalent causes for this problem [117].

The act of eating together “at the same table” is an important concept of commensality, well established among food studies scholars. This important human ritual provides benefits that goes beyond the biological need for food [118]. It is important to understand what happens beyond the food offered through the engagement with the material and affective elements of cooking and eating together, with special impact on social, isolation and loneliness. About 10% of the elderly in the UK report that they feel cut off from society and mainly say that the television is their main form of company.

The Christmas meal, as an exceptional commensality, must be analyzed through three main dimensions: first, eating together is in active communication between participants, regarded as interactional; second, “the staging of norms carried out by dinners and the control over those norms” is the normative dimension, and finally the symbolic dimension embracing the wide range of meanings attributed to eating together in different societies. Thus, the Christmas dinner may be analyzed in relation to several functions of establishing exchanges, social hierarchies and a sense of belonging to a group, which can unite and consolidate relationship, but also offer an occasion for differentiating and excluding social groups [118].

The lack of studies on this topic that include the geriatric population is another aspect we faced in trying to include a range of the adult population as large as possible. There is a limited number of studies that talk about the involvement of the elderly at the Christmas table, the women who prepare various types of food and the significance of the whole family on Christmas Eve [119]. However, there are no clear data related to caloric intake or the number of meals.

We consider it opportune to create a line of research in this direction, since the elderly constitute a growing segment of the world’s population, and syndromes such as frailty, depression along with social isolation influence the evolution of the elderly, especially at these times of the year.


**Tips and tricks**


1.Indulging in a special celebratory meal is unlikely to have a significant impact on your overall health. However, if you find yourself having multiple celebratory meals, it may be beneficial to adjust your other meals. For example, you could opt for a lighter evening meal, possibly you can choose to consume smaller portions of food or only one course at a festive meal.2.To complete a meal, whether you are dining out or at home, consider enjoying a cup of coffee or tea while others indulge in a dessert. Alternatively, you can choose to eat a green appetizer before the meal, for example a salad, which will bring you a rich supply of fiber, which will reduce your sweet tooth, or if you also choose dessert, the glycemic curve will be flatter, so your sweet tooth will be less in the future and you can skip dessert at your next meal [120].3.When drinking alcoholic beverages at home, it is easy to lose track of your consumption, so it is important to make a conscious effort to keep track of how much you are drinking because drinks add extra calories. In a study of 3327 men aged 60 to 79 years, without a history of myocardial infarction, stroke or diabetes, from general practices in 24 UK cities, men who consumed > or = 21 units/week showed higher levels of central adiposity than occasional drinkers or nondrinkers, regardless of the predominant type of beverage consumed (wine, beer, spirits, or mixed). The highest correlation was most clearly observed in beer and spirits consumers. It seems that the positive association was maintained regardless of whether the alcohol was drunk with or separate from the meals [121]. However, even non-alcoholic drinks can be high in calories, so, if possible, choose sugar-free beverages or water.4.Exploring alternative activities with your loved ones can be a great way to avoid unhealthy food options and even save money. Instead of solely focusing on meals, you can suggest going for a winter walk together. An interesting idea is the one addressed in a pilot randomized controlled trial in which 107 inactive adults from the UK aged between 18 and 75 were included through social media platforms, workplaces and community groups. Participants received an email with a Christmas-themed physical activity idea to complete that day. Each physical activity idea was presented in three intensity formats, including Easy Elf (light intensity), Moderate Mrs. Claus (moderate intensity) and Strenuous Santa (vigorous intensity). This type of activity not only encourages physical activity, but also provides an opportunity to enjoy quality time with family and friends in a healthy and fun way [122].5.Many times during holidays, high-calorie snacks are served between meals. If you feel the need to get a boost of energy between meals, opt for a salty snack that will not cause spikes in your glucose levels. For example, you can choose a Greek yogurt with a handful of pecans, some baby carrots with a spoonful of hummus or apple slices with cow’s cheese [120].6.The winter evenings spent in front of the TV watching our favorite movies with the family tempts us to eat on the couch and this makes us unable to keep track of the calories consumed [15]. Eating while watching TV can lead to increased food intake and a possible explanation lies in the multidimensional nature of distraction. It has been argued that once distracted from internal cues such as hunger and satiety by various external factors, an individual will eat mindlessly and their food intake will not be coded in certain ways which influences their desire to eat. The healthiest way is to eat at the table while enjoying the food [123].7.In winter, stores tempt us with hundreds of sparkling and colorfully packaged products at reduced prices, but you must be careful. Check the fat, sugar and number of calories on food labels when shopping and preparing food. Imagine how hard it would be to burn calories from a dessert rich in sugar and fat! [15].

Results from a study that sought to determine whether adding a menu chart that includes both the calorie content of the item and the amount of physical activity required to burn the calories in each item will lead the shopper to purchase a meal with fewer calories from a sample menu showing a significant difference in the average number of calories ordered by menu type.

8.In general, respondents ordered the lowest-calorie meals when shown the menu with calorie information and the number of miles they would walk to burn those calories. Those who were shown the menu with information about calories and the number of minutes of walking to burn those calories also chose the lower-calorie meals, although to a lesser extent. Pairwise comparisons revealed a statistically significant difference in the total number of calories ordered from the menu with miles walked to burn those calories compared to the menu without nutritional information [124]. Therefore, choose wisely and read the labels, including when shopping for the holidays. The holiday season can be stressful because people want the house to be perfectly decorated, the food to be tasty and the gifts to please everyone. This can trigger long-term emotional problems associated with loneliness, anxiety, and depression. In stressful times such as these, the consumption of comfort foods that are rich in calories, fat and sugar is common. For example, when rats were presented with a choice of highly palatable food such as lard or sugar, stress consistently increased intake of palatable food specifically. Humans similarly turn to hyperpalatable comfort foods such as fast food, snacks, and calorie-dense foods even in the absence of hunger and lack of homeostatic need for calories [125]. Therefore, it is important to keep stress levels under control during the holidays with sports, yoga meditation, and deep breathing [126].9.A lack of sleep leads to metabolic and hormonal imbalances such as decreased glucose tolerance, decreased insulin sensitivity, increased evening concentrations of cortisol, increased levels of ghrelin, decreased levels of leptin, and increased hunger and appetite leading to higher calorie intake. Even if the holiday season is busy, sleep as much as your body needs. In a study from 2011, it was discussed that approximately 50 epidemiological studies carried out in different geographical regions examined the association between sleep and obesity in adults and children. Most of them showed a significant association between short sleep (generally <6 h per night) and increased risk of obesity. A meta-analysis of 18 studies of 604,509 adults demonstrated a pooled obesity odds ratio (OR) of 1.55 (1.43–1.68; *p* < 0.0001) for less than 5 h of sleep and a dose effect of sleep duration so that for each additional hour of sleep BMI decreased by 0.35 kg/m^2^ [127].10.To make everything simpler, look for a friend to team up with to be motivated and accountable over the holidays!

## 5. Strengths and Limitations

In this study, the authors gathered information related to the impact of food during the winter vacation on the adult population. The unique approach is a strong point of the work, considering that we did not limit ourselves only to the studies presented, but also chose to present the possible psychological or consumerist reasons behind overeating at this time of the year and also information published during the COVID19 pandemic. A note of originality is given by the Tips and Tricks section, but also by the information related to physical activity. Among the limitations of this study, we can include the fact that the search for articles was performed on a single platform and we focused on a period of only 10 years.

## 6. Conclusions

Sustaining long-term behavioral changes and effectively managing obesity necessitates ongoing attention and support. Even the most effective short-term interventions are likely to produce continued positive outcomes with persistent intervention and support. It is essential to acknowledge that obesity is a multifaceted condition that requires a comprehensive and multidisciplinary approach to address underlying factors and provide ongoing assistance to individuals in their weight management endeavors.

Behaviors that have been linked to long-term success in weight loss include various factors. These imply regularly monitoring and weighing oneself, reducing calorie intake, consuming smaller and more frequent meals or snacks throughout the day, increasing physical activity, more at-home meals rather than restaurant or fast-food meals, limiting screen time, and incorporating portion-controlled meals or meal substitutes into the diet. All these measures should always be respected, including during the holidays.

## Figures and Tables

**Figure 1 nutrients-15-04201-f001:**
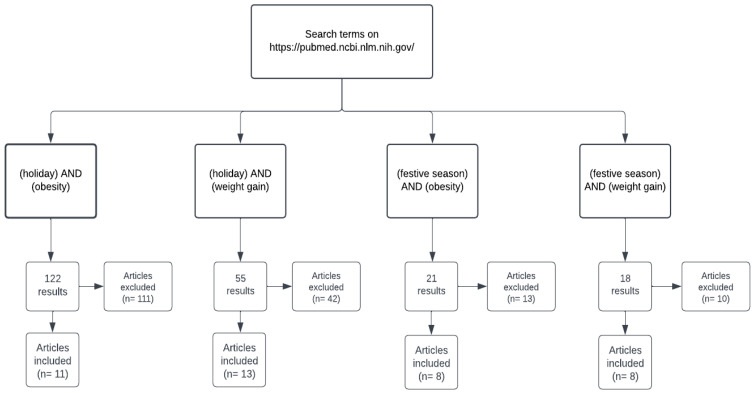
Flow diagram showing the selection process.

**Table 1 nutrients-15-04201-t001:** General characteristics of the studies included in the systematic review.

Author, Year, and Country	Study Duration	Number of Participants	Age, Sex,	BMI, Prevalence of Overweight/Obesity	Date of Measurements	Mean Weight Changes ± SD
Helander EE et al. (2016); USA, Germany, and Japan [10]	Thanksgiving (USA), Christmas (Germany), Golden Week (Japan); 1 year (2–6 months)	2924 (1781 USA, 760 Germany, 383 Japan);	USA: 42.4 years, 34% F. Germany 42.9 years, 34% F Japan: 41.6 years, 26% F.	USA: 24% obese; Germany:19% obese; Japan: 11% obese	1 August 2012–31 July 2013	Significant increases in weight pre- and post-Christmas across all three countries (0.4% in the USA; 0.6% in Germany; 0.5% in Japan). Significant weight increases occurred over Thanksgiving in the USA (0.2%), Golden Week in Japan (0.3%), and Easter in Germany (0.2%). Annual weights increased by 0.7% (0.6 kg) in the USA and 1.0% (0.8 kg) in Germany over the Christmas-New Year period, and 0.7% (0.5 kg) in Japan over Golden Week.
Cooper and Tokar, 2016, USA [11]	Between the months of March and August.	122	Mean age 32.2 ± 13.0 years 65% women (*n* = 79) and 35% men (*n* = 43),	57 normal weight—46.7%, 46 overweight—37.7%, 10 obese—15.6%, average BMI 25.8 ± 0.3 kg/m^2^	1 week before vacation, 1 week after vacation and 6 weeks after vacation	People with normal weight accumulating 0.28 ± 0.13 kg, while overweight people gained 0.39 ± 0.14 kg, and obese people gained 0.48 ± 0.27 kg
Bhutaniet al., 2020, Madison Metropolitan area [12]	15 September–15 January	23	Age 21–50 years	BMI: 30–39.9 kg/m^2^	15–30 September to 9–25 November (three visits) 9–25 November to 4–15 January (three visits)	A decrease of 0.86 kg during the pre-holiday period vs. an increase of 0.41 kg during the holiday period. Non-significant change with estimated energy intake in the expected direction (+80 kcal/day in holiday period vs. pre-holiday period)
Viñuela, et al., 2023, University of Castilla-La Mancha [13]	23 December–13 January	67, 10 men (14.9%) and 57 women (85.1%)	Mean age 19.00 ± 1.63 years for the males and 20.61 ± 4.33 year	BMI measurement (21.7 ± 3.1) and the second (21.9 ±3.1) and between the second and third (21.7 ± 3.9), observing a decrease	Four measurements: 23 December, 30 December, 6 January, 13 January	Over the Christmas holiday period, the overall weight tended to increase, from an initial weight of 59.6 ± 10.7 kg to 60.2 ± 10.6 kg at the end of the period, whereas 1 week after returning to university, the mean weight was 59.6 ± 9.9 kg. An increase between the first BMI measurement (21.7 ± 3.1) and the second (21.9 ± 3.1) and between the second and third (21.7 ± 3.9), observing a decrease in the return to university (22.6 ± 5.9) to values that were almost the same as at the start of the period but with a slight increase.
Olson, et al., 2020, USA [14]	November 2018–January 2019	683	54.6 years [SD: 13.2]	69% female, 93% white, BMI: 26.9 kg/m^2^ [SD: 5.5]	November 2018, January 2019	Participants gained 0.66 kg (SD: 1.85) from pre- to post-holiday and reported using an average of 12/18 strategies. More strategies were associated with less weight gain (F [1, 670] = 4.28). Daily self-weighing and prioritizing food choices were individually associated with less weight gain.
Mason et al., 2018, Birmingham, UK [15]	November 2016–February 2017	272 (136 were randomized to a brief behavioral intervention and 136 to a leaflet on healthy living)	Mean age: 43.9 years	BMI of ≥20 kg/m² 78% women, 22% men	Baseline assessments were conducted in November and December, with follow-up assessments in January and February (4–8 weeks after baseline)	The mean weight change was −0.13 kg (95% confidence interval −0.4 to 0.15) in the intervention group and 0.37 kg (0.12 to 0.62) in the comparator group. The adjusted mean difference in weight (intervention− comparator) was −0.49 kg (95% confidence interval −0.85 to −0.13). The odds ratio for gaining no more than 0.5 kg was nonsignificant.
Kaviani, et al., 2019, USA [16]	November–April	111	18–65 years	BMI ≥ 18.5 kg/m²	V1: before Thanksgiving, V2:after New Year’s Day, and the follow-up visit V3: 14 weeks after V2	There was no change in weight with DSW + GF, whereas the control group gained weight from v1 to v2 (−0.13 ± 0.27 kg vs. 2.65 ± 0.33 kg), respectively. In the control group, weight change was similar between individuals with overweight or obesity (OW/OB) vs. individuals with normal weight (2.71 ± 0.48 kg vs. 2.62 ± 0.43 kg, not significant, respectively). For DSW + GF, individuals with OW/OB lost weight, whereas those with normal weight-maintained weight during the holidays (−1.46 ± 0.62 kg vs. 0.33 ± 0.27 kg, respectively). The control group lost weight during the follow-up (−1.14 ± 0.43 kg; v2 to v3) but retained 57% of weight gain; therefore, weight gain from v1 to v3 was significant (1.51 ± 0.39 kg).
Wilson et al., 2019, USA [17]	2015–2016	239 (100 employees in year 1 (2015–16) and 139 employees in Year 2. 36 repeated participants	47.1 years (+10.46)	Mean weight: 196.7 lb/89.2 kg. Approximately 90% of the participants were female. A majority of respondents were African-American (71%), followed by white (24%) and others (5%)	End of October to mid-January (weight was measured every two weeks during the program (at 2, 4, 6, and 8 weeks) as part of the intervention)	During the program, participants lost a significant amount of weight (from 196.7 lb/89.2 kg to 192.3 lb/87.2 kg), losing weight at each weigh-in. To examine weight maintenance between programs, data were analyzed from the 36 repeat participants (who participated in both years of the program). Participants, on average lost 6.9 lb (3.1 kg) in Year 1, gained 8.7 lb (4.0 kg) between the programs, and lost 4.3 lb (2.0 kg) in Year 2 of the program resulting in a net loss of 2.5 lb (1.1 kg)
Stevenson, et al., 2013 [18]	57 ± 0.5 days (2013)	148	Age 18–65 years	48 males and 100 females, mean body mass index of 25.1 ± 0.5 kg/m²	Mid-November (visit 1) and early January (visit 2)	Participants showed significant increases in BW (0.78 ± 0.1 kg), BF% (0.5 ± 0.2%,) systolic blood pressure (SBP; 2.3 ± 1.2 mm Hg), and diastolic blood pressure (1.8 ± 0.8 mm Hg. Obese participants (35.2 ± 0.8 kg/m^2^) showed a greater increase in BF% compared with normal weight participants (21.7 ± 0.2 kg/m^2^) and a trend vs. overweight participant (26.8 ± 0.3 kg/m^2^). Exercise (4.8 ± 0.6 h per week) did not protect against holiday weight gain and was not a significant predictor for changes in BW or BF%.
Ramirez-Jimenez, et al., 2020, Barcelona, Spain [19]	November–January	38 (TRAIN group, *n* = 16, HOLID group, *n*= 22)	57 ± 8 years	BMI 32 ± 5 kg/m² and metabolic syndrome	20 December (visit 1) and 10 January (visit 2)	HOLID group increased body weight (91.3 ± 13.0 to 92.0 ± 13.4 kg), mean arterial pressure (94.0 ± 10.6 to 97.1 ± 8.9 mmHg, blood insulin (10.2 ± 3.8 to 12.5 ± 5.4 µIU·mL^−1^) and HOMA (3.2 ± 1.3 to 4.1 ± 2.3). In contrast, TRAIN prevented those disarrangements and reduced total (170.6 ± 30.6 to 161.3 ± 31.3 mg·dL^−1^) and low-density lipoprotein cholesterol (LDL-C, 104.8 ± 26.1 to 95.6 ± 21.7 mg·dL^−1^.

**Table 2 nutrients-15-04201-t002:** Nutritional information of traditional Christmas foods.

Country	Dishes (100 g)	Proteins	Carbohydrate	Fat	Calories	Citation
UK	Christmas pudding	5 g	49.3 g	7.1 g	279	[33]
	British roast turkey	26 g	4 g	2.1 g	116	[34]
	Prawn cocktail	6.5 g	21.5 g	1.2 g	125	[35]
France	Pate de foie gras	11.4 g	4.67 g	43.8 g	462	[36]
	Scallops in orange-butter sauce	14.7 g	7.1 g	8 g	163.2	[37]
	Smoked salmon mousse	12 g	6 g	20.9 g	248	[38]
Greece	Roasted lamb	23.93 g	-	18.15 g	266	[39]
	Dolmadakia	3.5 g	23.8 g	4 g	599	[40]
	Christopsomo	10.9 g	18.1 g	8 g	308.2	[41]
Italy	Panettone	6 g	45 g	13 g	320	[42]
	Cannoli	8.7 g	28.8 g	11 g	254	[43]
	Cassata	2.5 g	23.1 g	9.5 g	182	[44]
	Artigianale gelato al pistacchio	5 g	24.4 g	9.6 g	204	[45]
	Buccellato	5.1 g	66.7 g	11.3 g	397	[46]
	Tiramisu	4.7 g	34.1 g	16.4 g	317	[47]
Spain	Sopa de galets	11 g	72 g	1.5 g	354	[48]
	Polvoron	9.2 g	55.5 g	23.3	470.3	[49]
	Mantecados	5.7 g	54.2 g	31.4 g	525.7	[50]
	Roscon de reyes	5.4 g	45 g	14 g	333	[51]
Norway	Pinnekjøtt	29.8 g	-	37 g	452.2	[52]
	Swede and carrot mash	0.8 g	11.5 g	3.3 g	83	[53]
	Ribbe	13.6 g	-	34.7 g	367.8	[54]
	Lutefisk	5.71 g	0.5 g	0.1 g	25	[55]
	Aquavit (100 mL)	-	-	-	244	[56]
Germany	Feuerzangenbowle (100 mL)	-	14 g	-	127	[57]
	Roasted duck	18.9 g	-	28.2 g	336	[58]
	Sausage stuffing	5.6 g	24.2 g	8.8 g	195.7	[59]
	Potato dumplings	4.2 g	27.5 g	2.8 g	153.3	[60]
	Red cabbage	1.2 g	8 g	-	46	[61]
	Stollen	5 g	7.7 g	63.2 g	336.7	[62]
USA	Prime rib	25.9 g	-	17.3 g	266	[63]
	Cranberry sauce	-	-	44.7 g	186.7	[64]
	Cornbread	6.7 g	42.2 g	9.4 g	282	[65]
	Mashed potato	1.8 g	15.7 g	3.5 g	100	[66]
	Sweet potato casserole	2 g	32 g	3 g	160	[67]
	Gingerbread cookies	4,7 g	71.4 g	14,2 g	428.5	[68]
Japan	Strawberry shortcake	2.3 g	18.5 g	10.1 g	171.5	[69]
	Cream stew	5.2 g	8.6 g	4.6 g	94.9	[70]
	Creamy Japanese potato salad	6 g	17 g	14 g	215	[71]
Romania	Pork cabbage	8.4 g	10.7 g	12.2 g	186.2	[72]
	Sweet bread (Cozonac)	9 g	51 g	5.1 g	298	[73]
	Smoked ham	16.9 g	-	35 g	388	[74]
	Salty bacon	3.9 g	-	85 g	781	[74]
	Leberwurst	17.5 g	0.7 g	24.2 g	302	[75]
	Tobă	23 g	-	22 g	299	[75]
	Țuică 25%	9 g	-	-	175	[74]
	Wine 8%	-	-	-	60	[74]
Colombia	Bunuelos	8 g	48.6 g	26.2 g	426	[76]
	Empanadas	11.3 g	31.2 g	18.4 g	335	[77]
	Ajiaco Colombiano	5.1 g	5.3 g	2.3 g	62.2	[78]
	Colombian sancocho	3.71 g	4.9 g	9.2 g	117.5	[79]
	Natilla	3.9 g	17.6 g	4 g	122	[80]
	Aguardiente (100 mL)	-	-	-	222	[81]

## Data Availability

Data supporting this article are included within the reference list.

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
