# Peer review of "Winter Holidays and Their Impact on Eating Behavior—A Systematic Review"

_nutrients, 2023, doi:10.3390/nu15194201_

Round 1

Reviewer 1 Report

The study's authors focused on the effect of the holidays on the eating patterns of adults. The paper is well-constructed and clear. The discussion was carried out correctly. The presented conclusions are consistent. The tables are well described. References typical and actual. 

Shortcomings are listed below: 

-      There is a lack of citations in the lines 54 and 58.

-      Change “(9)” to “[9]” in the line 90.

-      Name the section including discussion "Discussion" –line 177 and below

-      Add "Study Limitation" section.

-      Check the punctuation and spelling in the whole text.

Minor editing is needed - punctuation and spelling errors.

Author Response

Dear reviewer,

Thank you for all your comments.

  1. We have added citations in the lines 54 and 58. Thank you!
  2. We have made the changes from (9) to [9].
  3. We have named the section Discussion.
  4. We have added a Study Limitation section.

Hope we have touched all the points you asked us to change.

If there are any other changes you consider we should make, please let us know.

Yours sincerely,

All the authors

Reviewer 2 Report

This review aimed to evaluate the effect of winter holidays on eating patterns and weight fluctuations among adults. The study analyzed 10 articles involving a total of 4,627 participants. The results showed that during the holiday period, particularly around Christmas and New Year, individuals tend to experience weight fluctuations and increased caloric intake. The study observed weight gains ranging from 0.2% to 0.6% across different countries during holiday periods. Notably, around half of the weight gained during the holidays was maintained even after the study period, highlighting the potential long-term impact of holiday-related weight gain.

The authors concluded that overeating during the holidays is a widespread phenomenon influenced by social norms, gatherings, and the desire for a fresh start in the new year. The study emphasized the importance of understanding these patterns for developing targeted interventions and prevention programs to address holiday-related weight gain.

In summary, the systematic review provides insights into the impact of winter holidays on adults' eating habits and weight fluctuations, emphasizing the need for strategies to mitigate the effects of holiday overeating.

1. there is plagiarism, in particular in the table, the outcomes of the articles should be summarised not just reported

2. The paper's findings appear to state the obvious, as the phenomenon of weight gain during holiday periods is widely recognized and hardly requires an extensive systematic review.

3. The rationale behind the study seems quite self-evident and doesn't offer any groundbreaking insights, as it's common knowledge that people tend to overeat during holidays and that such behaviors contribute to weight gain.

4. While the research effort is appreciated, dedicating a systematic review to demonstrate what is already widely acknowledged might not provide substantial value to the field of nutrition research.

Suggestions: 

Narrower Focus and Novel Insights: Instead of a broad review of the obvious, the authors could narrow their focus. They could explore specific holiday periods, cultural differences in holiday eating habits, or delve into the psychological reasons behind overeating during celebrations. This could provide more unique and valuable insights.

In-depth Analysis: Rather than simply stating the well-known connection between holidays and weight gain, the authors could analyze different types of holiday foods, their nutritional compositions, and how they contribute to the observed weight changes. A more detailed analysis would add depth to their findings.

Comparative Study: To bring more substance to the paper, the authors could compare holiday-related weight gain with weight gain during other times of the year. This could involve a meta-analysis of existing studies or the authors' data collection to highlight any significant differences and contributing factors.

Behavioral Patterns: Instead of just acknowledging the weight gain, the authors could investigate the underlying behavioral patterns that lead to overindulgence. This could involve surveys, interviews, or even experiments to understand why people make certain food choices during holidays.

Practical Implications: Move beyond the obvious by offering practical suggestions to mitigate holiday weight gain. Whether it's through mindful eating tips, creating healthier holiday recipes, or suggesting post-celebration fitness routines, providing actionable advice can significantly improve the paper's relevance.

ENGLISH IS FINE

Author Response

Dear reviewer,

Thank you for all your comments.

  1. We have focused mostly on the Christmas period. We have added a table that includes specific foods eaten in different countries at Christmas and calories for each of them. Thank you!
  2. We have added studies that evaluated cultural differences and beliefs in the Christmas period.
  3. We have added a comparative study that compared holiday-related weight gain with weight gain during other times of the year. Thank you for your recommendation.
  4. We have added a section that includes behavioral patterns that lead to overindulgence.
  5. We have also added some practical implications that can be made during the Christmas period in order to diminish weight gain. Thank you!

Hope we have touched all the points you asked us to change.

If there are any other changes you consider we should make, please let us know.

Yours sincerely,

All the authors

Round 2

Reviewer 2 Report

The paper's premise, which emphasizes that greater energy consumption during holidays correlates with weight gain, aligns with well-established knowledge in the field of nutrition. The link between indulgent holiday eating habits and subsequent weight gain is widely recognized, making the rationale behind this review self-evident.

The authors revieweed mainly the discussion. The discussion in the paper provides a comprehensive overview of the factors contributing to weight gain during the holiday season and includes data from various studies. It highlights several key points:

Overeating During the Holidays: The discussion begins by acknowledging that overeating during the Christmas and New Year period is common and challenging to modify. Social norms, gatherings with loved ones, and the desire to start the new year with healthier habits are mentioned as contributing factors.

Caloric Intake During Holidays: The paper cites a study by Elina E. Helander and colleagues, which tracked weight fluctuations in participants from multiple countries. It notes that excessive caloric consumption, often three times the recommended daily intake, is a significant contributor to holiday weight gain.

Long-Term Effects: The discussion emphasizes that approximately half of the weight gained during holiday periods is maintained even after the holidays are over, suggesting that holiday-related weight gain can have lasting effects on individuals' weight.

Weight Gain in Relation to BMI: The paper references studies indicating that overweight or obese individuals are more prone to gaining weight during holidays compared to those of normal weight. This suggests a potential risk of an "obesity epidemic" over time.

Impact of Caloric Intake and Dining Out: The correlation between increased caloric intake, especially during dining out, and holiday weight gain is highlighted, indicating that dining habits during holidays contribute significantly to weight fluctuations.

Social and Environmental Factors: The discussion acknowledges that social factors, such as holiday celebrations with high-calorie foods, and environmental factors, like larger portion sizes, play a role in encouraging overeating during the holiday season.

Traditional Christmas Foods and Calories: The paper provides insights into traditional Christmas foods and their calorie content in various countries, shedding light on cultural influences on holiday eating habits.

Overall, the discussion effectively presents a detailed analysis of the reasons behind holiday weight gain and provides data from multiple studies to support its findings. It highlights the importance of understanding these factors for developing effective prevention strategies and interventions to address holiday-related weight gain.

Despite the evidence presented in the debate on the tendency to gain weight during the holidays, it is important to emphasise that the basic rationale seems to be quite obvious. Increased calorie consumption during the holidays, often accompanied by a decrease in physical activity due to festivities and holidays, can naturally lead to weight gain.

However, these studies provide concrete and detailed data quantifying the extent of weight gain and highlight the fact that such an increase can have long-term effects on people's health, especially those who are overweight or obese. Therefore, despite the obvious link between overeating during the holidays and weight gain, these studies provide a scientific basis for better understanding the phenomenon and developing more targeted prevention strategies.

Three suggestions the authors could consider to improve their work:

1. Delving into Psychological Causes: The authors could devote a specific section to understanding the psychological causes of weight gain during the holidays. This could include an analysis of the desire to indulge during the holidays, the stress associated with celebrations and the role of social expectations in eating behaviour. A deeper understanding of the psychological reasons could enable the development of more targeted prevention strategies.

2. Include Prevention Strategies: The authors could add a section on prevention strategies for weight gain during the holidays. They could discuss practical approaches that people can adopt to maintain a balanced diet and active lifestyle during this period. These strategies could be evidence-based and could include advice on how to manage overeating and promote physical activity.

3. Consider Multidisciplinary Approaches: The authors could further explore the idea of involving experts from different disciplines to address the problem of weight gain during the holidays. For example, they could discuss how mental health professionals, nutritionists and personal trainers could collaborate to offer comprehensive support to people trying to maintain their weight during the holidays.

English is fine

Author Response

Dear reviewer,

Thank you for all your comments.

  1. We have added a section entitled “Deeper understanding of psychological reasons”. Thank you for your suggestion.
  2. We have added a section entitled “Prevention strategies for weight gain”.
  3. We have added a section entitled “Tips and tricks” that includes useful multidisciplinary tips that are easy to implement.

Hope we have touched all the points you asked us to change.

If there are any other changes you consider we should make, please let us know.

Yours sincerely,

All the authors

Round 3

Reviewer 2 Report

The authors have made significant improvements to the manuscript, enhancing its clarity and depth. The paper provides a comprehensive overview of the challenges and strategies related to weight management during the holiday season. The use of multiple studies to support the arguments adds credibility to the paper.

However, there are some areas that could benefit from further refinement:

Study Limitations: While the authors acknowledge the limitations, including the focus on a single platform for article search and the 10-year time frame, it would be beneficial to discuss the implications of these limitations on the study's findings.

Specificity in Tips and Tricks: The "Tips and Tricks" section is helpful but could be more impactful with specific examples or case studies to illustrate each point.

Inclusion of Extreme Age Groups: The paper does not include populations with extreme age, such as children and the elderly. Given that holiday eating habits can significantly impact these groups, their inclusion could provide a more holistic view.

Clarification on Physical Activity: The paper presents conflicting views on the role of physical activity in weight management. While it's stated that exercise may not prevent weight gain, it's also suggested as a strategy for weight management. This could be confusing for the reader and may benefit from further clarification.

Psychological Factors: The paper touches upon stress and emotional eating but could delve deeper into the psychological factors affecting holiday weight gain, perhaps suggesting coping mechanisms.

Citations and References: Ensure that all cited studies are appropriately referenced, and consider including a broader range of sources to strengthen the paper's arguments. The section discussing changes in eating habits could benefit from a broader perspective. It might be useful to explore recent research that investigates how extraordinary events, such as pandemics, can impact regular eating patterns.Similarly, the discussion on reduced physical activity could be enriched by considering studies that focus on its effects on individuals with high cardiovascular risk, especially during periods of restricted movement like lockdowns.

Overall, the paper is well-structured and informative but could be enhanced by addressing these points.

English is fine

Author Response

Dear reviewer,

Thank you for all your comments.

  1. We have added more details in the study limitation section. Thank you!
  2. In the “Tips and tricks” section, we have included more specific examples or case studies to illustrate each point.
  3. Regarding the extreme age groups, we included a section regarding the geriatric population. As for the children, we did not include them from the beginning in our study, this fact being exposed in the materials and methods section. We truly believe that children evaluation can be studied in a different systematic review, do to their complexity (starting from the parents and their eating habits and finishing with their age, group or even cultural behavior). Thank you for your remark!
  4. We have made clarifications regarding the physical activity, as you mentioned. Thank you!
  5. We have added more details regarding the psychological factors, including coping mechanisms.
  6. We have added more citations and references, as you mentioned.

Hope we have touched all the points you asked us to change.

If there are any other changes you consider we should make, please let us know.

Yours sincerely,

All the authors